# Characterization of the Gaseous and Odour Emissions from the Composting of Conventional Sewage Sludge

**Daniel González [1,2], Nagore Guerra [3], Joan Colón [3]**, **David Gabriel [2]**, **Sergio Ponsá [3] and Antoni Sánchez [1,\*]**

[1] Composting Research Group (GICOM), Department of Chemical, Biological and Environmental Engineering, Universitat Autònoma de Barcelona, 08193 Bellaterra, Spain; daniel.gonzalez.ale@uab.cat

[2] Group of Biological Treatment of Liquid and Gaseous Effluents (GENOCOV), Department of Chemical, Biological and Environmental Engineering, Universitat Autònoma de Barcelona, 08193 Bellaterra, Spain; David.Gabriel@uab.cat

[3] BETA Technology Centre: "U Science Tech", Univeristy of Vic-Central University of Catalonia, 08500 Vic, Spain; nagore.guerra@uvic.cat (N.G.); joan.colon@uvic.cat (J.C.); sergio.ponsa@uvic.cat (S.P.)

**\*** Correspondence: antoni.sanchez@uab.cat

**Abstract:** Many different alternatives exist to manage and treat sewage sludge, all with the common drawback of causing environmental and odour impacts. The main objective of this work is to present a full inventory of the gaseous and odorous emissions generated during the bench-scale composting of conventional sewage sludge, aiming at assessing the process performance and providing global valuable information of the different gaseous emission patterns and emission factors found for greenhouse gases (GHG) and odorant pollutants during the conventional sewage sludge composting process. The main process parameters evaluated were the temperature of the material, specific airflow, average oxygen uptake rate (OUR), and final dynamic respiration index (DRI), resulting in a proper performance of the sewage sludge composting process and obtaining the expected final product. The obtained material was properly stabilized, presenting a final DRI of $1.2 \pm 0.2$ g $O_2 \cdot h^{-1} \cdot kg^{-1}$ Volatile Solids (VS). GHGs emission factor, in terms of kg $CO_{2eq} \cdot Mg^{-1}$ dry matter of sewage sludge (DM–SS), was found to be $2.30 \times 10^2$. On the other hand, the sewage sludge composting odour emission factor (OEF) was $2.68 \times 10^7$ ou$\cdot Mg^{-1}$ DM–SS. Finally, the most abundant volatile organic compounds (VOC) species found in the composting gaseous emissions were terpenes, sulphur compounds, ketones, and aromatic hydrocarbons, whereas the major odour contributors identified were dimethyldisulphide, eucalyptol, and $\alpha$-pinene.

**Keywords:** sewage sludge; composting; gaseous emission; greenhouse gases (GHG); odour; VOCs

## 1. Introduction

The large amount of sewage sludge (SS) production is becoming a worldwide environmental problem. In Europe, the increasing number of wastewater treatment plants due to the application of the Council Directive 91/271/EEC concerning urban wastewater treatment [1] has led to the generation of more than 7.5 million Mg of SS per year (in dry basis) by the different European Member States [2], with the existing necessity of handling and managing it. Nowadays, different treatment processes are used to handle and manage all SS produced, such as land application, incineration, composting, or anaerobic digestion, aiming at reducing its volume and quantity as well as stabilizing it [2]. In fact, one of the most typical and applied stabilization approach, composting, is considered an environmentally friendly technology that reduces sludge volume and transport costs, decomposing organic matter into a stable end product and removing malodorous compounds as well as pathogens and weeds [3].

The aim of SS composting is obtaining a biological stable product that can be used as fertilizer or soil amendment by decomposing the organic matter present in the initial feedstocks [4]. Nevertheless, different unavoidable environmental and social concerns arise from this SS biological treatment, such as the emission of different gaseous compounds responsible for odour nuisance and atmospheric pollution. In this sense, ammonia, hydrogen sulphide, and a wide range of volatile organic compounds (VOCs) have been described as the main odorant compounds present in SS composting gaseous emissions [5,6], which can appear at different levels due to the biological degradation of organic matter and nitrogen-and sulphur-based compounds depending on the conditions (aerobic/anaerobic) of the SS composting process. Apart from the previously cited odorant compounds, different greenhouse gases (GHG) such as carbon dioxide, methane, and nitrous oxide are emitted during the SS composting process. From those, even though $CO_2$ is the principal product of the biological respiration emitted during SS biodegradation, it is not accounted into the total amount of GHG in composting processes due to its biogenic origin [7]. Moreover, $CH_4$ and $N_2O$ present a higher global warming potential (GWP), 28 and 265 times that of $CO_2$, respectively [8].

Due to its importance in environmental impact assessment and the valuable information that it represents when it comes to proposing mitigation strategies and avoiding population nuisance, gaseous emission of GHG and odorant compounds has been widely reported for SS composting in pilot-scale [6,9], and some studies have been conducted in full-scale facilities [10,11]. Emission factors referring to specific pollutants are a useful tool for environmental impact assessment and for estimating the overall emission rate of a treatment process based on a specific activity index, which should represent the process evaluated [12,13]. The US Environmental Protection Agency (USEPA) defined the emission factor of a chemical compound as the relation between its concentration and the emitted air flow and then normalized with respect to one or more reference parameters such as the mass of waste to be treated, the emitting surface, or time units amongst others [14]. Analogously, in the last years this tool has been referenced for the assessment of odour impact from different treatment processes and facilities, namely the odour emission factor (OEF) [15–17].

Odour emissions can be characterized using high-performance analytical techniques for the identification and quantification of odorous compounds such as gas chromatography coupled to mass spectrometry (GC/MS). However, it is difficult to relate the information obtained from those techniques with the real sensation perceived by humans. In this sense, sensorial analytical techniques such as dynamic olfactometry are frequently used for odour impact assessment purposes. Their use helps into quantifying odours in terms of intensity and/or concentration [18]. Moreover, complementary information about the odorous emissions generated during the SS composting process can be obtain by means of the individual odour activity values ($OAV_i$), which can help in identifying the main odour contributors present in a complex gaseous mixture by relating the concentration of a specific odorant compound measured in a complex gas mixture with its odour detection threshold (ODT).

The main objective of this work is to provide a full and global inventory of the gaseous and odorous emissions generated during the conventional sewage sludge composting process, which could be later extrapolated to higher process scale. The authors intend to enlarge the scientific knowledge about gaseous and odorous emissions generated during the composting of SS due to its importance for further development of the composting technology and the development of mitigation and control strategies. Emission factors for specific pollutants are provided, as well as a screening on the diversity of VOC families found in the gaseous emissions from both treatment technologies, which can be valuable information for further environmental assessment studies.

## 2. Experiments

### 2.1. Characteristics of the Feedstocks

SS was obtained from the wastewater treatment plant (WWTP) of Navàs, Barcelona, which treats 1500 $m^3 \cdot d^{-1}$ of domestic wastewater. Diatomaceous earth (DE) was obtained from the organic fraction

of municipal solid wastes (OFMSW) biomethanization plant of Can Barba in Terrassa, Barcelona, and pruning waste (PW), used as bulking agent, was obtained from the composting plant of Manresa, Barcelona. The mixing ratio of the three feedstocks was 1:0.25:3 (SS:DE:PW) on a volume basis (1:0.15:0.48 *w/w* basis), which is a typical mixing ratio used in SS composting [10]. Table 1 shows the physical–chemical characteristics of each material and mixture.

**Table 1.** Physical–chemical properties of the feedstocks and mixture used during the sewage sludge (SS) composting process.

| Physical–Chemical Properties | SS | PW | DE | Composting Mixture |
|:---:|:---:|:---:|:---:|:---:|
| Moisture content (%) | 83.5 ± 0.1 | 18.0 ± 0.0 | 3.9 ± 0.0 | 59.2 ± 0.0 |
| Organic matter content (%) | 68.9 ± 0.2 | 88.7 ± 0.0 | 35.2 ± 0.1 | 85.9 ± 0.0 |
| C/N ratio | 9.1 | 31.6 | 1321.6 | 13.6 |
| pH | 6.7 | 8.2 | 6.2 | – |
| Electrical conductivity (mS·cm$^{-1}$) | 2.5 | 1.7 | 1.2 | – |
| DRI$_{24\,h}$ (g O$_2$·h$^{-1}$·kg$^{-1}$ VS) | 7.3 ± 0.9 | – | – | 6.3 ± 1.7 |

PW—pruning waste; DE—diatomaceous earth; DRI—dynamic respiration index.

*2.2. Sewage Sludge Composting Biorreactor Operation*

Figure 1 shows the bioreactor with cylindrical structure and operative volume of 100 L that was used to carry out the SS composting process. The reactor was filled with 43.15 kg of the mixture with a density of 395.9 kg·m$^{-3}$. The reactor was 0.85 m high and had a diameter of 0.50 m. The internal and external walls of the reactor were made of stainless steel, and they had a thermal isolation of polyurethane foam (2 cm) to avoid the loss of temperature and to maintain adiabatic conditions. A perforated plate was fixed at the bottom of the reactor to support the material and to facilitate aeration, working as a diffusor. The composting reactor was hermetically closed. Temperature of the material was measured with a thermometer probe (PT-100, Iserntech S.A., Vilanova i la Geltrú, Spain) introduced into the composting reactor, with sensors located at the top, middle, and bottom of each one. An air compressor (Dixair DNX 2050, Worthington Creyssensac, Pinto, Spain) and gas mass-flowmeters/controllers (D-6311-DR, Bronkhorst High-Tech B. V., Ruurlo, the Netherlands) were used for continuous aeration. Aeration flow rate was controlled by the oxygen uptake rate (OUR) [19], a composting key parameter, which determines the oxygen consumption and the level of biological activity present in the solid matrix. To follow the evolution of the composting process in terms of biodegradation, the oxygen content of the outlet gases of the reactor was monitored using an O$_2$ sensor (O2 A1, Alphasense, Great Notley, Essex, UK), after passing them through a condensation train to prevent sensors being damaged. The composting process lasted 20 days, and the material was mixed once on the 10th day. Arduino ONE was used for data acquisition, and LabView 2017$^{®}$ was used for data analysis and airflow control.

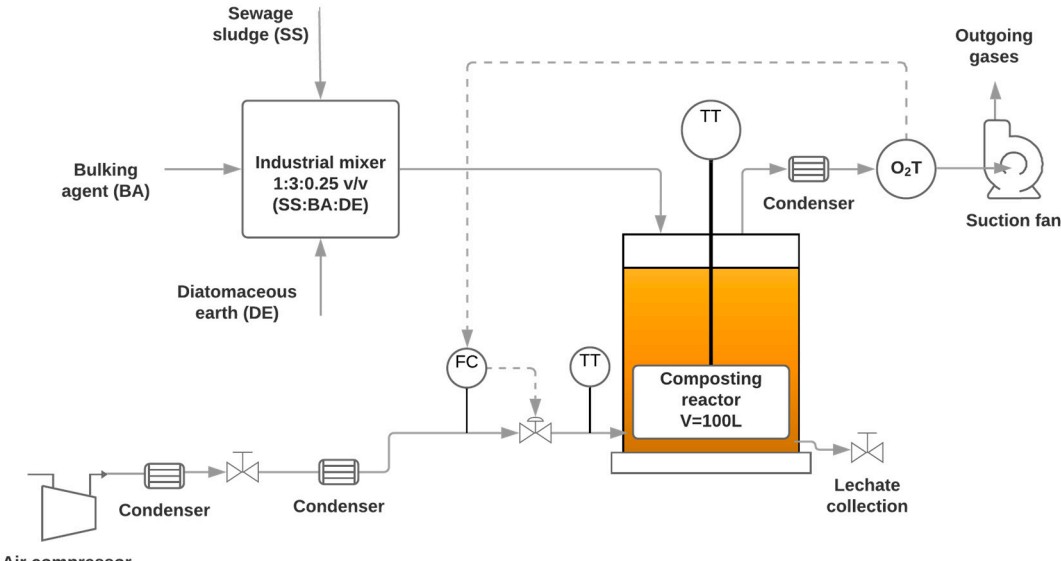

**Figure 1.** Schematic of the bench-scale composting reactor.

## 2.3. Gas Sampling Methodology

For the composting reactor, gas sampling was done directly from the outlet gas tube to obtain the gaseous samples into Nalophan® bags. All Nalophan® bags used to store gas samples were flushed two times before obtaining the final gaseous sample for analysis.

To perform VOC characterization, two different gaseous samples of 1 L each were prepared in sorbent tubes packed with two different hydrophobic sorbents (Tenax® TA and Carbograph™ 1TD, Markes International, Inc., Gold River, CA, USA) by means of a manual sampling pump (Markes International, Inc.). The gaseous samples were obtained on the 2nd and 11th day of running, coinciding with the thermophilic and the mesophilic phases of the SS composting process.

## 2.4. Analytical Methodology

Moisture content, dry matter, organic matter content, N-Kjeldhal, C/N ratio, pH, and electrical conductivity were determined according to the standard procedures [20,21]. The experimental measure of OUR was determined from online data acquired on airflow and oxygen content [19,22]. To evaluate the final biological stability of the composted material, the dynamic respiration index (DRI) was used. The determination of the DRI was done using a dynamic respirometer [23] as explained in [10].

Methane analysis was performed using an Agilent 6890 N gas chromatograph (GC) (Agilent Technologies, Inc., Santa Clara, CA, USA), equipped with a photoionization flame detector (PID). The column used for the analysis was a HP-PLOT Q semi-capillary column (30 m × 0.53 mm × 40.0 μm, Agilent Technologies, Inc., Santa Clara, CA, USA), with $N_2$ as carrier gas at 2 psi pressure, and a post-column particle trap (2 m, no. 5181–3352, Agilent Technologies, Inc., Santa Clara, CA, USA). The injector temperature was 240 °C, the detector temperature was 250 °C and the oven, which worked isothermally, was at 60 °C. The injection volume used for each sample was 500 μL and the total time of analysis was 4 min.

Nitrous oxide analysis was performed using an Agilent 6890N GC (Agilent Technologies, Inc.), equipped with an electron capture detector (ECD). The column used for the $N_2O$ analysis was a HP-PLOT Q semi-capillary column (30 m × 0.53 mm × 40.0 μm, Agilent Technologies, Inc.), with $N_2$ as carrier gas at 2 psi pressure, and a post-column particle trap (2 m, 5181–3352, Agilent Technologies, Inc.). The injector temperature was 120 °C, the detector temperature was 345 °C, and the oven, which worked isothermally, was at 60 °C. The injection volume used for each sample was 500 μL and the total time of analysis was 6 min.

Total volatile organic compounds (tVOCs), ammonia, and hydrogen sulphide concentration of the outlet gases of the composting reactor were measured daily in situ with a MultiRAE Lite portable analyser (RAE Systems, San José, CA, USA), equipped with a 10.6 eV PID lamp for tVOCs measurement and two electrochemical sensors for $NH_3$ and $H_2S$ measurement, respectively. tVOCs detection ranged from 0 to 1000 $ppm_{veq}$ isobutylene with 1 $ppm_v$ increments, $NH_3$ detection ranged from 0 to 100 $ppm_v$ with 1 $ppm_v$ increments, and $H_2S$ detection ranged from 0 to 100 $ppm_v$ with 1 $ppm_v$ increments. During each analysis, the portable analyser was placed inside a hermetic recipient with inlet and outlet ports and the outlet gases of the reactor were passed through it until a steady value was read on the analyser. For the composting reactor, the outlet gas flowrate was the same as the inlet airflow as the reactor was hermetically closed. The gases were measured just before the water trap installed to protect the rest of the measurement devices from the gaseous flow moisture.

An SM-100 portable field olfactometer provided by Scentroid (IDES Canada, Inc., Whitchurch-Stouffville, ON, Canada) was used to perform the odour concentration analysis of the different gaseous samples obtained during the experiment, as explained by [24]. Briefly, the portable field olfactometer, which is factory calibrated, works by diluting a fixed flow of clean breathable air with the odorous sample until the panellist perceives an odour change. Both the clean air and the odorous sample flows are mixed by means of a Venturi valve, and the dilution ratio is controlled by means of a manual screw and different interchangeable measurement plates, depending on the expected odour concentration. All samples were analysed in triplicate by the same panellist in a closed, well-ventilated room the same day of sampling.

To perform the VOCs' characterization, first a liquid VOC custom solution was prepared with 35 standard compounds (considered as representative VOCs in gaseous emissions from composting processes) in methanol, for calibration purposes. To analyse the obtained samples, first a UNITY-2 thermal desorber (TD) (Markes International, Inc., Gold River, CA, USA) was used to desorb the gaseous samples retained in the sorbent tubes. These sorbent tubes were heated at 290 °C for 8 min while flowing high-purity He at a flow rate of 50 mL·min$^{-1}$ to desorb the VOCs onto a cold trap at −10 °C. Then the cold trap was heated up to 305 °C at a rate of 40 °C·s$^{-1}$ for 5 min to desorb the VOCs and to inject them into the chromatographic column, using a 1:10 split ratio to prevent column overload. The gas was directed to the chromatographic column through a transfer line heated at 250 °C to avoid condensation. Then, an Agilent 7820 GC coupled to an Agilent 5975 mass spectrometer (MS) (Agilent Technologies, Inc.) was used to analyse and characterize the different VOCs present in the gaseous samples obtained at different moments of the process. The chromatographic column used was an Agilent DB-624 capillary column (60 m × 0.25 mm × 1.4 μm, Agilent Technologies, Inc.), with an He gas flow rate of 1 mL·min$^{-1}$ as the carrier gas. The temperature program for the GC was an initial isothermal stage at 50 °C during 2 min, then a first temperature ramp to 170 °C at a rate of 3 °C·min$^{-1}$, followed by a second ramp up to 280 °C at a rate of 8 °C·min$^{-1}$. Finally, the MS acquired data in scan mode with *m/z* interval ranging from 35 to 355 amu. The VOCs were identified by mass spectra matching with the Wiley275 mass spectrum library available in the TD-GC/MS system. The gaseous samples were analysed the same day that they were obtained in order to preserve the stability of VOCs in sorbent tubes [25].

*2.5. Estimation of the Emission Factors*

The emission factor calculation was based on the measured concentration of a specific pollutant, the aeration flow through the reactor and, in this case, it was normalized by the total mass of dry matter of sewage sludge (DM–SS) introduced in the reactor. The first step was to obtain the daily emission rate of each pollutant (Equation (1)).

$$ER_i = C_i \cdot F, \tag{1}$$

where $ER_i$ is the emission rate of the target pollutant (mg pollutant·d$^{-1}$); $C_i$ is the measured concentration of the target pollutant (mg pollutant·m$^{-3}$); F is the aeration flow through the reactor (m$^3$·d$^{-1}$).

Then, the emission rates of a target pollutant obtained for each sampling day were represented versus process time, so the area below the curve obtained corresponds to the emitted total mass of the pollutant throughout the composting process studied. Finally, the emitted total mass of a target pollutant was divided by the total DM–SS introduced in the reactor to obtain the emission factor (Equation (2)).

$$EF_i = ER_i/I,\tag{2}$$

where $EF_i$ is the emission factor of the target pollutant (mg pollutant·kg$^{-1}$ SS); $ER_i$ is the emission rate of the target pollutant (mg pollutant·d$^{-1}$); I is the total DM–SS treated as specific activity index (kg DM–SS).

### 2.6. Potential Odour Contributors

The individual odour activity value ($OAV_i$) can be calculated as the relation between the concentration of a target compound and its odour detection threshold, as described by Equation (3) [26,27]. $OAV_i$ is an indicator of the contribution to odour of a specific quantified compound present in a gaseous sample containing a mixture of odorant compounds [28,29]. In this sense, the higher the $OAV_i$, the more likely an odorant compound contributes to the perceived odour.

$$OAV_i = C_i/ODT_i,\tag{3}$$

where $OAV_i$ is the individual odour activity value (dimensionless); $C_i$ is the measured concentration of the target pollutant (ppb$_v$); $ODT_i$ is the odour detection threshold of the target compound (ppb$_v$).

## 3. Results and Discussion

### 3.1. Sewage Sludge Composting Performance

The aim of the SS composting process is to obtain a biologically stable product with high agronomic properties that is mature enough, to be used as fertilizer or soil amendment, by biological decomposition of the organic matter [30]. Figure 2 shows the performance of the composting process during the 20 day run.

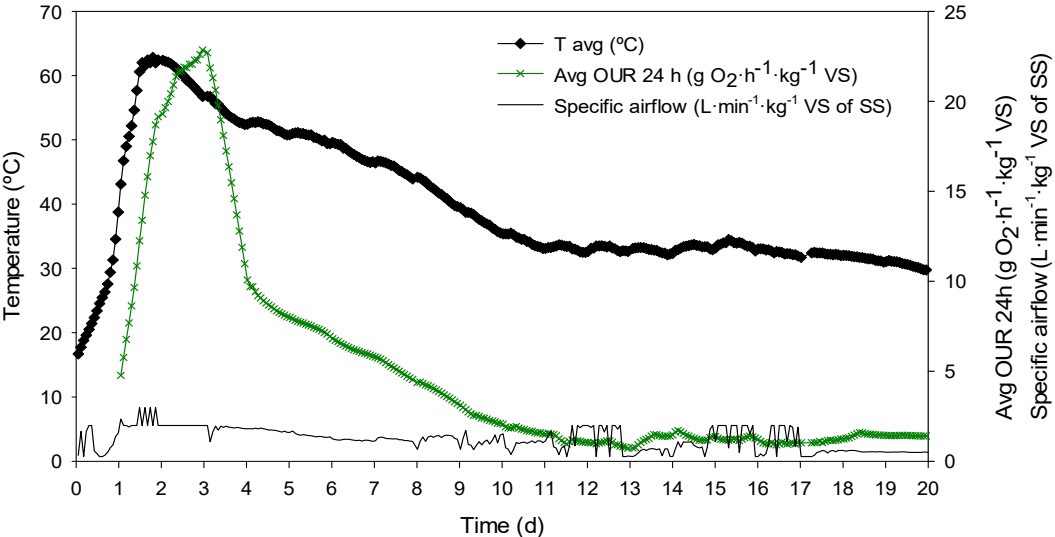

**Figure 2.** Performance of the composting process. Black dots represent the evolution of the average temperature of the reactor (°C); green crosses represent de average oxygen uptake rate (OUR) value for 24 h (g O$_2$·h$^{-1}$·kg$^{-1}$ VS); solid line represents the specific airflow passed through the reactor (L·min$^{-1}$·kg$^{-1}$ VS of SS).

In Figure 2, the evolution of the average temperature of the bottom, middle, and top of the solid matrix; the specific airflow; and the average OUR of the last 24 h are represented for the whole process time of the composting treatment. As expected, during the composting process, two temperature stages can be differentiated, a thermophilic stage (>40 °C) and a mesophilic stage (<40 °C). Temperature increased rapidly during the first 48 h to the maximum value registered (62.9 °C). Thermophilic conditions were reached at the 24 h of process and maintained until the 9th day of composting. The composting reactor was turned just once on the 11th day of the process. However, despite the turning, the temperature tended to decrease and never reached the peak temperature again.

During the composting process, the OUR was monitored to control the aeration rate of the reactor and to verify the proper performance of the biological process. Specific airflow supplied to the reactor varied from 2.9 to 0.3 L·min$^{-1}$·kg$^{-1}$ VS of SS during the composting process, with an average value of 1.2 L·min$^{-1}$·kg$^{-1}$ VS of SS, which is typical of values used during SS composting [6,31]. As it can be observed in Figure 2, the curve described by the OUR corresponds to a typical SS composting process [6,32], where the peak of maximum biological activity in terms of organic matter biodegradation is found during the 2nd and 3rd day of the process, coinciding with when the maximum temperature is reached inside the reactor. Then, the tendency of the OUR was to decrease as easily degradable organic matter is being consumed. According to the temperature and OUR profiles and the DRI value of the final material (1.2 ± 0.2 g O$_2$·h$^{-1}$·kg$^{-1}$ VS), which was reduced an 81% with respect to the mixture's initial DRI, the composting was performed properly and the material was stabilized. The SS composting process time was fixed at 20 days on the basis of following similar procedures that have been generally described by the GICOM research group in different SS composting studies [6,10,22]. Moreover, the fact that non-digested SS was used, which is a waste that presents higher biological activity, was a reason to establish this duration in order to ensure a proper biological stabilization of the treated mixture.

### 3.2. Greenhouse Gases Emission

Greenhouse gases (GHGs) emissions from different SS composting processes have been widely reported in the past years due to their potential impact to global warming [31,33,34]. Figure 3 shows the daily emission rates of CH$_4$ and N$_2$O obtained during the composting process, altogether with the temperature profile of the material and the specific airflow supplied to the reactor at the moment of sampling.

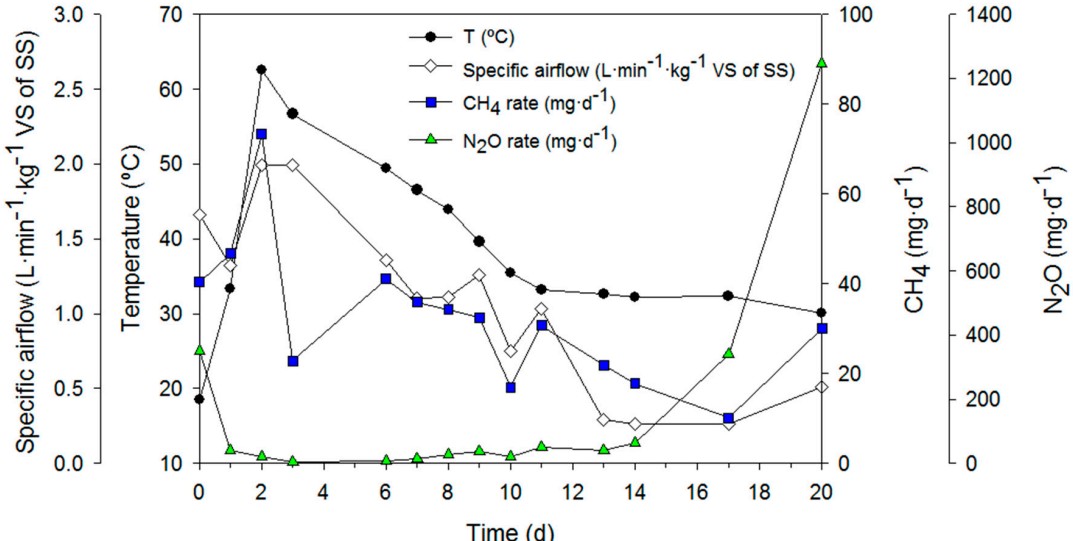

**Figure 3.** CH$_4$ and N$_2$O emission rate profiles during the composting process. Black dots represent the average temperature of the reactor (°C); white diamonds represent the specific airflow passed through the reactor (L·min$^{-1}$·kg$^{-1}$ VS of SS); blue squares represent the emission rate of CH$_4$ (mg CH$_4$·d$^{-1}$); green triangles represents the emission rates of N$_2$O (mg N$_2$O·d$^{-1}$).

Two different emission dynamics can be observed for $CH_4$ and $N_2O$ in Figure 3. First, $CH_4$ emission followed a similar profile as the reactor's temperature profile, presenting its emission peak when maximum temperature and maximum aeration occurred. Differently, $N_2O$ showed an emission peak on the first sampling day due to the stripping of some remaining $N_2O$ present in the SS. Then, the $N_2O$ emission rapidly decreased to low values, possibly due to the inhibition of some mechanisms of $N_2O$ generation by the high temperature achieved by the system [35]. Later, a great increase of the $N_2O$ emission was observed when the temperature of the material was below 40 °C.

Generally, $CH_4$ is formed by methanogens under anaerobic conditions. Hence, because of the rapid degradation of organic matter during thermophilic phase, an anaerobic environment can be created by the oxygen demand at this temperature range [31]. As observed in Figure 3, the peak of methane emission coincided with that of temperature, and then $CH_4$ emission decreased as temperature decreased into the mesophilic range. The maximum $CH_4$ concentration measured during SS composting was 27.01 $ppm_v$, whereas the minimum concentration obtained was 1.34 $ppm_v$.

On the other hand, the formation and emission of $N_2O$ is complex, which can be produced by denitrification, when nitrates are converted to $N_2O$ and nitrogen gas, and incomplete nitrification, when during the conversion of ammonia to nitrite there is some formation of $N_2O$ [36]. As it can be observed in Figure 3, during the first sampling day, a peak of $N_2O$ emission appeared, mainly because of the stripping effect of the remaining $N_2O$ present in the SS used for the composting experiment. Once this first $N_2O$ was stripped of the SS, low levels of $N_2O$ formation and emission were observed until the temperature of the material decreased to a mesophilic range. At that point, an increase of the $N_2O$ emission appeared until the end of each process. The results suggest that the observed $N_2O$ formation has a direct relation with the material's temperature at the moment of the emission [35]—below 40 °C—and with the low carbon availability at the latter stages of the process. At these conditions, some denitrifying bacteria can be reactivated and promote the $N_2O$ formation by nitrate denitrification at the end of the processes [35,37,38], which could be confirmed by further bacterial community characterization in future works [39]. Moreover, the increase in the $N_2O$ emission coincide with low aeration of the material. This fact possibly promoted the formation of anaerobic spots around the composting material and hence, the higher formation and emission of $N_2O$ during the last phase of the process. Peaks of $N_2O$ emission have been reported in different composting studies [35,40,41] mainly in non-aerated composting piles, when carbon availability was low, which supports that a combination of the previously mentioned conditions (low temperature, low C availability, and low oxygenation) can be the reason for the observed $N_2O$ increase at the end of the present SS composting process. Maximum $N_2O$ concentration measured during SS composting was 317.59 $ppm_v$, whereas the minimum concentration obtained was 0.28 $ppm_v$.

Table 2 shows the emission factor of each GHG for the SS composting process, expressed in kg of $CH_4$ or $N_2O$ per Mg DM–SS and in kg of $CO_{2eq}$ per Mg DM–SS.

**Table 2.** $CH_4$ and $N_2O$ emission factors, expressed in kg of pollutant per Mg of dry matter of sewage sludge (DM–SS) and in kg of $CO_2$ equivalent per Mg of dry matter of SS.

| | |
|---|---|
| $CH_4$ emission factor (kg $CH_4 \cdot Mg^{-1}$ DM–SS) | $1.37 \times 10^{-1}$ |
| $N_2O$ emission factor (kg $N_2O \cdot Mg^{-1}$ DM–SS) | $8.53 \times 10^{-1}$ |
| $CH_4$ emission factor (kg $CO_{2eq} \cdot Mg^{-1}$ DM–SS) [a] | 3.83 |
| $N_2O$ emission factor (kg $CO_{2eq} \cdot Mg^{-1}$ DM–SS) [a] | $2.26 \times 10^2$ |
| GHG emission factor (kg $CO_{2eq} \cdot Mg^{-1}$ DM–SS) [a] | $2.30 \times 10^2$ |

[a] Greenhouse gas (GHG) emission for $CH_4$ and $N_2O$, on a 100-year frame, is 28 and 265 times higher than that of $CO_2$, respectively [8].

To contextualize the obtained results, the GHG emission factors have been contrasted with other similar studies. For example, [31] reported a maximum GHG emission factor of 284.13 kg $CO_{2eq} \cdot Mg^{-1}$

DM–SS during the composting of dewatered non-digested sewage sludge in a 60 L aerated pilot-scale reactor. On the other hand, [6] found lower GHG emission factors referred to the SS composting in a 50 L aerated pilot-scale reactor, ranging from 4.98 to 9.90 kg $CO_{2eq} \cdot Mg^{-1}$ DM–SS. Another bench-scale composting study conducted with dewatered sludge with wheat straw in a 150 L reactor [34] reported GHG emission factors ranging from 36.05 to 134.56 kg $CO_{2eq} \cdot Mg^{-1}$ DM of mixture. Generally, a high variability exists in terms of GHG emission depending on different factors such as the characteristics of the feedstocks, the operational parameters (aeration strategy) or even the scale of the process. In the present study, the fact that $N_2O$ emission increased sharply at the end of the SS composting process contributed to a higher GHG emission factor. A shorter SS composting process would not be the best solution to minimize $N_2O$ emission at the end of the process because the biological stabilization of the material could be compromised. In this sense, a better aeration strategy or an increase of the mixing frequency at the latter stages of the process to avoid or at least minimize the formation of anaerobic spots around the composting material could have been beneficial to mitigate $N_2O$ emissions and, hence, to decrease GHG emissions.

### 3.3. Odorant Compounds Emission

### 3.3.1. $NH_3$, $H_2S$, tVOCs and Odour Emission

The emission of $NH_3$, $H_2S$, tVOCs, and odour from the SS composting treatment was monitored during the whole process. Table 3 presents the emission factors obtained for all these pollutants, expressed in kg of pollutant per Mg DM–SS and ou per Mg DM–SS, respectively, when monitoring the composting process.

**Table 3.** $NH_3$, $H_2S$, tVOCs and odour emission factors, expressed in kg of pollutant per Mg of dry matter of SS and in ou per Mg of dry matter of SS, respectively.

| | |
|---|---|
| $NH_3$ emission factor (kg $NH_3 \cdot Mg^{-1}$ DM–SS) | 5.13 |
| $H_2S$ emission factor (kg $H_2S \cdot Mg^{-1}$ DM–SS) | $1.66 \times 10^{-1}$ |
| tVOCs emission factor (kg C-VOC$\cdot Mg^{-1}$ DM–SS) | 6.20 |
| Odour emission factor (ou$\cdot Mg^{-1}$ DM–SS) | $2.68 \times 10^7$ |

To put the obtained results into context, the emission factors obtained have been compared with other reported emission factors from different composting processes. Researchers in [6] reported $NH_3$ and tVOCs emission factors in the range of 1.33–4.12 kg $NH_3 \cdot Mg^{-1}$ DM–SS and 0.66–1.00 kg C-VOC$\cdot Mg^{-1}$ DM–SS, respectively, during composting of raw sludge in a 50 L aerated pilot-scale reactor. González et al. [10] studied the emission of $NH_3$ and tVOCs in a full-scale sewage sludge composting plant, reporting emission factors of 25.32 kg $NH_3 \cdot Mg^{-1}$ DM-SS and 6.17 kg C-VOC$\cdot Mg^{-1}$ DM-SS. Maximum concentrations measured during SS composting were 2600 ppm$_v$ of $NH_3$, 66 ppm$_v$ of $H_2S$, and 1650 ppm$_v$ of tVOCs, which were observed during the peak of maximum temperature of the reactor. On the other hand, few works dealing with the OEFs of SS biological treatment have been found. However, [42] evaluated the OEFs of different substrates in a 300 L aerated pilot-scale reactor, reporting an OEF of $9.35 \times 10^8$ ou$\cdot Mg^{-1}$ DM-SS for SS composting and [24] obtained an OEF of $9.45 \times 10^7$ ou$\cdot Mg^{-1}$ DM-SS in a full-scale sewage sludge composting plant. For SS composting, different ranges of target pollutants and odour concentrations can be found in literature depending on the characteristics of the feedstocks or the composting process itself [43–45].

Figure 4 shows the daily odour emission rates (OER) obtained during the SS composting process, together with the average temperature of the reactor and the specific airflow.

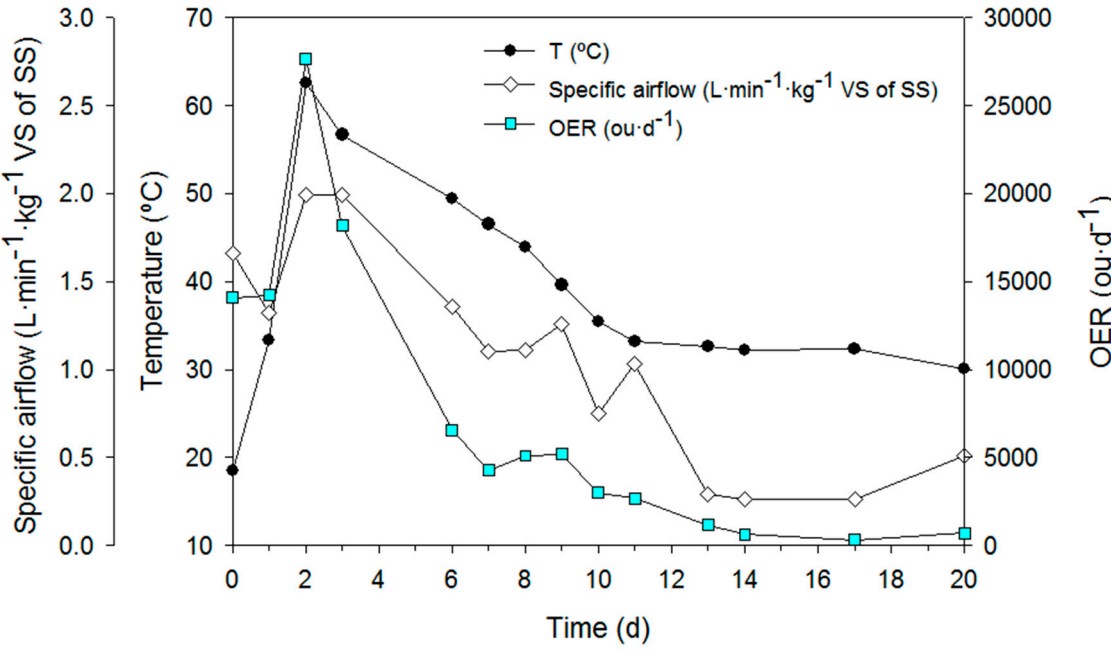

**Figure 4.** Odour emission rate (OER) profile during the composting process. Black dots represent the average temperature of the reactor (°C); white diamonds represent the specific airflow passed through the reactor (L·min⁻¹·kg⁻¹ VS of SS); cyan squares represent the odour emission rate (ou·d⁻¹).

Similarly to other works where odour emission from SS composting were studied [24,42], the profiles of odour emission reached its maximum during the thermophilic phase, when biological activity is higher and then tended to decrease as temperature and biological activity decreased. In fact, the distribution of the odour emissions during the whole SS composting process shows that more than 64% of these emissions happened during the thermophilic phase of the process, whereas the rest was divided between the pre-thermophilic and the mesophilic stages. Temperature and hence biological activity have been reported to be key factors affecting odour emission, being higher during the thermophilic phase of the composting process [24,46,47], due to the fact that easily biodegradable organic matter, as well as nitrogen-and sulphur-based compounds, are consumed, leading to the formation and emission of higher quantities of odorant compounds such as ammonia, hydrogen sulphide, and a wide range of VOCs. Maximum odour concentration measured during composting was 3200 ou·m⁻³. As the processes went on, odour emission decreased progressively until reaching low odour concentration values between 280 and 290 ou·m⁻³.

### 3.3.2. VOCs' Characterization

A chemical characterization was made by comparing the VOC families found in the different samples as well as by quantifying the concentration of 17 out of the 35 detected compounds. Figure 5 shows the distribution of the different VOC families found in the samples of two different days obtained during the SS composting process.

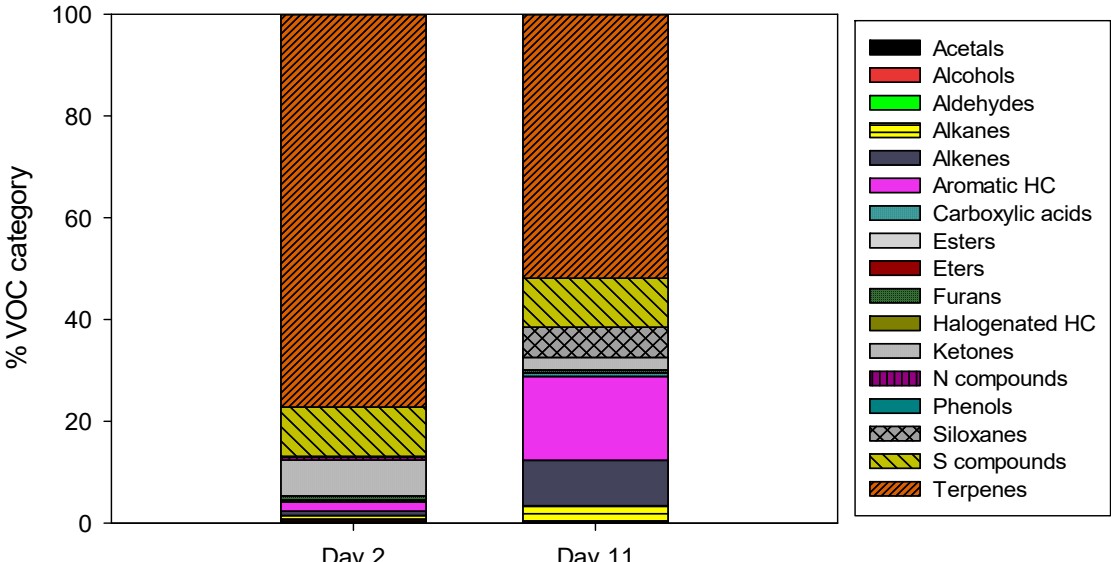

**Figure 5.** Distribution of the different volatile organic compound (VOC) categories found in the off gases of the composting reactor, expressed in relative abundance with respect to the whole sample.

As it can be observed in Figure 5, terpenes—predominantly α-pinene—are the main VOC family found in the gaseous samples analysed, with relative abundances between 50–80%. Furthermore, other odorant species appeared in different percentages, such as sulphur compounds—dimethyl sulphide (DMS) and dimethyl disulphide (DMDS), ketones—mainly 2-butanone and 2-pentanone, aromatic hydrocarbons—with toluene and m-xylene, and some alkanes. Generally, terpenes such as α-pinene, β-pinene, or limonene may originate as intermediate products of aerobic biodegradation of organic matter. Furthermore, terpenes are also related to the PW used as bulking agent for the composting process [24,48,49]. Particularly in this case, its abundance was that high due to the high SS:PW *v/v* ratio used. On the other hand, the higher abundance of sulphur compounds over other VOC families in both samples is normal as those compounds are typical chemicals present in the gaseous emissions generated during the SS handling and treatment [29,49]. The relative abundances of the different species found in the samples of the 2nd day of each process were similar, with nearly 80% of terpenes, 10% of sulphur compounds, and 7% of ketones among others. However, there was a variation in the relative abundances of the different VOC families found in the samples of the 11th day; terpenes were still the predominant species in the gaseous samples, but decreased to 50%, and sulphur compounds were maintained about 9%, while the abundance of other species such as aromatic hydrocarbons and alkenes increased.

There exists a clear relation on how the concentration of different individual VOCs can contribute to the odour concentration of a complex gas mixture. In this sense, the individual odour activity value (OAV$_i$) of the VOCs quantified in the gaseous samples obtained from the sewage sludge composting process was determined, as it can be a useful tool to indicate how an individual compound contributes to odour in a gaseous mixture of odorant compounds [28]. The calculated OAV$_i$ are presented in Table 4 altogether with the concentration and the odour detection threshold of each specific VOC.

**Table 4.** VOCs quantified in the gaseous emissions generated during the sewage sludge composting process (concentration expressed in $ppb_v$), with its odour detection threshold (ODT, $ppb_v$) and the calculated $OAV_i$ ("<DL": compound concentration below the detection limit; "−": compound not detected).

| Family | Compound | ODT | VOC Concentration | | $OAV_i$ | |
|---|---|---|---|---|---|---|
| | | | Day 2 | Day 11 | Day 2 | Day 11 |
| Alkanes | Heptane | 670 [a] | 58.2 | <DL | $8.7 \times 10^{-2}$ | <DL |
| Aromatic HC | Benzene | 2700 [a] | 3.7 | <DL | $1.4 \times 10^{-3}$ | <DL |
| | Toluene | 330 [a] | 69.7 | <DL | 0.2 | <DL |
| | m-xylene | 0.041 [a] | 10.7 | <DL | 261.0 | <DL |
| | Styrene | 35 [a] | 254.9 | | 7.3 | - |
| Ketones | 2-butanone | 440 [a] | 5793.9 | | 13.2 | - |
| | 2-pentanone | 28 [a] | 2369.1 | | 84.6 | - |
| N compounds | Pyridine | 63 [a] | 910.6 | | 14.5 | - |
| | Indole | 0.3 [a] | 7.5 | | 25.0 | - |
| S compounds | DMS | 3 [a] | - | 70.9 | - | 23.6 |
| | DMDS | 2.2 [a] | 37,813.0 | 70.1 | 17,187.7 | 31.9 |
| Terpenes | $\alpha$-pinene | 18 [a] | 13,299.1 | 12.8 | 738.8 | 0.7 |
| | $\beta$-pinene | 33 [a] | 6390.3 | - | 193.6 | - |
| | Limonene | 38 [a] | 5490.0 | 2.3 | 144.5 | $6.1 \times 10^{-2}$ |
| | p-cymene | 1200 [b] | 1545.4 | 0.9 | 1.3 | $7.5 \times 10^{-4}$ |
| | Eucalyptol | 12 [c] | 13,604.8 | - | 1133.7 | - |

[a] [27]; [b] [50]; [c] [51].

More than half of the VOCs quantified in the gaseous samples exceeded their ODT and were found to be smell-relevant compounds, responsible for typical odour emitted during sewage sludge treatment. To define the VOCs that presented major contribution to odour found in both gaseous samples, the compounds with an $OAV_i$ higher than 1 have been presented in Figure 6, together with the odour concentration measured for those gaseous samples.

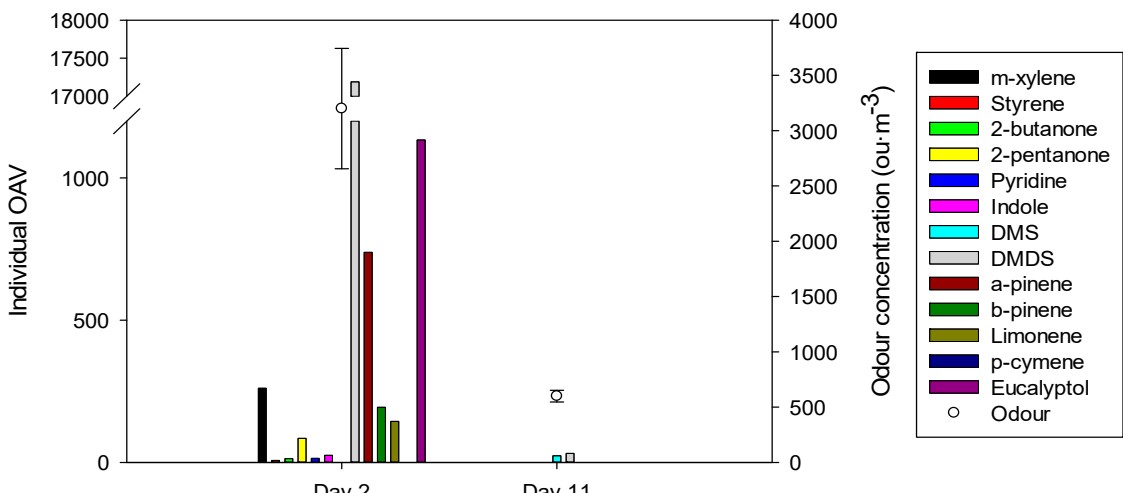

**Figure 6.** Odour concentration and individual odour activity value ($OAV_i$) >1 determined for the gaseous samples obtained during the thermophilic phase (day 2) and the mesophilic phase (day 11) of the conventional sewage sludge composting process.

As it can be observed in Figure 6, it is clear that DMDS is the major contributor to odour found in the gaseous emissions generated during the thermophilic phase (day 2) of the sewage sludge

composting process, which coincides with the maximum odour concentration measured during the whole process. Moreover, other compounds such as eucalyptol, α-pinene, or m-xylene also contributed to a large extent to odour emission generated during the thermophilic phase. On the other hand, during the mesophilic phase (day 11), odour emission decreased to very low levels around 600 ou·m$^{-3}$, and the VOCs' concentration and, hence, the OAV$_i$ decreased as well. Nevertheless, sulphur compounds such as DMS and DMDS were quantified and determined as the major odour contributor found during the mesophilic phase.

Even though in the present work there was not enough information to deal with that, some authors have attempted to correlate odour concentration with the numerical addition of the OAV (OAV$_{SUM}$) of all the individual compounds quantified in the emissions, seeking a relation between the concentration of odorants and odour concentration, obtaining uneven results. For example, [26] found a good linear correlation between the OAV$_{SUM}$ and odour concentration measured during composting of pig slaughterhouse sludge at lab-scale ($R^2 = 0.87$). However, [24,52,53] were unable to strongly correlate the odour concentration with OAV$_{SUM}$ in a full-scale sewage sludge ($R^2 = 0.60$), in a swine farm ($0.30 < R^2 < 0.52$) and in a sanitary landfill ($R^2 = 0.39$). These results point out that OAV$_{SUM}$ may not be a robust approach or a good odour concentration predictor when dealing with complex odorous mixtures because many different aspects can interfere, such as the type or characteristics of the waste to be treated, the existence of synergistic and masking effects between odorants found in the gaseous mixtures, the limitations that can be encountered when quantifying different odorants (lack of quantification of different odorant species), or the over- or under-estimation of odour concentration [54].

## 4. Conclusions

Results indicated a proper performance of the SS composting process, obtaining a biologically stable material. The GHGs emission (in terms of kg of CO$_{2eq}$) during the SS composting process were mainly dependant on the emission of N$_2$O due to its higher global warming potential. N$_2$O emission was found to greatly increase at the end of the process, coinciding with the decay of the material to a mesophilic temperature stage, together with the low aeration provided. On the other hand, odour emissions followed the temperature dynamics as expected, presenting their maximum during the thermophilic phase of the composting process. Regarding this, DMDS, eucalyptol, and α-pinene were found to be the major odour contributors present in the gaseous emissions. In conclusion, this study encloses a full inventory of the gaseous and odorous emission related to the SS composting process, which can help in the development and further implementation of mitigation and control strategies of the mentioned emissions.

**Author Contributions:** Conceptualization, D.G. (David Gabriel), A.S., J.C., N.G., and D.G. (Daniel González); methodology, D.G. (David Gabriel), A.S., J.C., N.G., and D.G. (Daniel González); sampling and analysis, N.G. and D.G. (Daniel González); writing—original draft preparation, D.G. (Daniel González); writing—review and editing, D.G. (Daniel González), N.G., D.G. (David Gabriel), A.S., J.C., and S.P.; supervision, D.G. (David Gabriel), A.S., J.C., and S.P. All authors have read and agreed to the published version of the manuscript.

**Funding:** This research received no external funding.

**Acknowledgments:** The authors would like to thank the cooperation and kindness of the wastewater treatment plant, the OFMSW biomethanization, and the composting plant authorities and working personnel and their interest in carrying out this study.

**Conflicts of Interest:** The authors declare no conflict of interest.

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
