# Peer review of "Characterization of the Gaseous and Odour Emissions from the Composting of Conventional Sewage Sludge"

_atmosphere, doi:10.3390/atmos11020211_

Round 1

Reviewer 1 Report

This manuscript presents a characterization of greenhouse gas and odorous pollutants emission from composting of sewage sludge. The study is interesting and provides new insight on the topic.

Firstly, I would like to receive some more explanation by the authors about the duration selected for conducting the experiment (20 days). Did this length depended on some parameters, or it was a “fixed” variable? It seems to me that the N2O emission during the last days (and consequently the high N2O emission factor) highly depends on the aeration. The authors mention about possible anaerobic spots at lines 268-270 (I agree with that), but this in fact affects the final result. Thus, I think that authors should better clarify about the duration of the experiment.

Another important comment is that the discussion of the results is poor, in particular for what concerns odor emissions. Results should be analysed under qualitative and quantitative aspects, and compared to results of OAV calculation from other studies on sewage sludge and composting (if available), (see for example https://doi.org/10.1007/s11356-019-06939-5).

Still regarding the analysis of odor emissions, the authors should i) specify the methods employed for the analysis (UNI, EPA etc.) in relation to the target compounds and ii) better describe the working principle of the field olfactometer adopted.

Regarding the analysis of GHG emission, as CO2 is not considered in the study, the authors should specify in the manuscript what assumptions for CO2 emissions were considered.

Finally, I recommend other minor corrections:

Figure 4 should be improved by reporting also VOC, H2S and NH3 emission. The layout of this figure should be improved, as the blue part hides some information. Figure 3 is not clear and should be revised, as the green area cover most of the information reported herein What about H2S? Its concentration was measured up to 66 ppmv, but it did not appear as a substance with high OAV (this is more of a personal curiosity seen that H2S has a very low OT); Acronyms should always be explained at their first appearance (e.g. OFMSW at line 90);

Author Response

Response to reviewer #1

The process time selection was done in accordance to typical lab-scale studies performed by other authors as introduced in the manuscript. Moreover, as the main objective of a SS composting process is the biological stabilization of the material, this time was necessary to reach proper stabilization values in terms of DRI (Page 8 Lines 259-263).

As it is observed in Figure 3, the emission of N2O sharply increased at the latter stages of the composting process for different reasons or, possibly, a conjunction of them. Low aeration, low carbon availability and low temperature are factors that have been described to affect N2O emissions, generally increasing them. In this sense, a shorter process would not be the best solution because biological stabilization of the material can be compromised, but for example a better aeration strategy or an increased mixing frequency could be beneficial to minimize the late N2O emissions. This discussion has been added to the manuscript (Page 10 Lines 305-309 & Lines 329-336).

Discussion regarding odour emissions has been improved. In the sense of OAV and the use of the sum of individual OAV (OAVSUM), we did not use this parameter because we found it unfair and not representative or robust enough to properly correlate it with odour concentration when dealing with complex odorous mixtures. Different authors have tried to correlate these two parameters obtaining uneven results. Many different aspects can affect this OAVSUM such as the type or characteristics of the waste to be treated, the existence of synergistic and masking effects between different odorants found in a complex gaseous mixture, the limitations that can be encountered when quantifying the different odorant compounds (lack of quantification, for example) or the over-underestimation of odour concentrations. Moreover, in our case, with just two samples for VOCs characterization is not possible to perform any analysis in terms of odour concentration and OAVSUM relationship (Page 12 Lines 371-379; Page 16 Lines 446-459).

The working principle of the portable field olfactometer has been briefly explained in the manuscript (Page 6 Lines 175-180), together with a precise description of the VOCs analysis methodology (Page 6 Lines 182-200)

Regarding the analysis of CO2, it was not accounted into the greenhouse gases inventory due to its biogenic origin. Moreover, as it has been added in the manuscript, the global Warming Potential of CH4 and N2O are much higher than that from CO2 (28 and 265 times, respectively) (Page 2 Lines 57-64).

Figures 3 and 4 were in SigmaPlot format, presenting a layout error when converting the manuscript to a pdf file. They have been corrected to avoid hiding information (Page 9 Line 271; Page 12 Line 364).

All the acronyms have been checked so they are now explained before their appearance.

Reviewer 2 Report

I recommend authors to indicate in the abstract for the abreviation GHG explication text for this abreviation. 

I recommend to change the title of this publication, from actual title:

A thorough characterization of the gaseous and odour emissions from the composting of conventional sewage sludge.

to:

Characterization of the gaseous and odour emissions from the composting of conventional sewage sludge.

Author Response

Response to reviewer #2

New information has been added to the introduction to give some more background on the problematic that arises from sewage sludge generation and management, the formation of odorant compounds during sewage sludge composting and greenhouse gases emissions (Page 2 Lines 39-42 & Lines 55-64).

All the abbreviations used in the manuscript have been revised so they are now explained before their appearance.

The title of the manuscript has been modified. Actual title is: “Characterization of the gaseous and odorous emissions from the composting of conventional sewage sludge”.